# WGS Revealed Novel *BBS5* Pathogenic Variants, Missed by WES, Causing Ciliary Structure and Function Defects

**DOI:** 10.3390/ijms24108729

**Published:** 2023-05-13

**Authors:** Adella Karam, Clarisse Delvallée, Alejandro Estrada-Cuzcano, Véronique Geoffroy, Jean-Baptiste Lamouche, Anne-Sophie Leuvrey, Elsa Nourisson, Julien Tarabeux, Corinne Stoetzel, Sophie Scheidecker, Louise Frances Porter, Emmanuelle Génin, Richard Redon, Florian Sandron, Anne Boland, Jean-François Deleuze, Nicolas Le May, Hélène Dollfus, Jean Muller

**Affiliations:** 1Laboratoire de Génétique Médicale, UMR_S INSERM U1112, Institut de Génétique Médicale d’Alsace (IGMA), Faculté de Médecine FMTS, Université de Strasbourg, 67000 Strasbourg, France; 2Laboratoires de Diagnostic Génétique, Hôpitaux Universitaires de Strasbourg, 67000 Strasbourg, Franceelsa.nourisson@chru-strasbourg.fr (E.N.);; 3Centre de Référence Pour les Affections Rares en Génétique Ophtalmologique (CARGO), Institut de Génétique Médicale d’Alsace (IGMA), Filière SENSGENE, Hôpitaux Universitaires de Strasbourg, 67091 Strasbourg, France; 4Inserm, Université de Brest, EFS, UMR 1078, GGB, F-29200 Brest, France; 5CHU Nantes, CNRS, INSERM, L’institut du Thorax, Nantes Université, 44000 Nantes, France; 6CEA, Centre National de Recherche en Génomique Humaine, Université Paris-Saclay, 91057 Evry, France; 7Service de Génétique Médicale, Institut de Génétique Médicale d’Alsace (IGMA), Hôpitaux Universitaires de Strasbourg, 67000 Strasbourg, France; 8Unité Fonctionnelle de Bioinformatique Médicale Appliquée au Diagnostic (UF7363), Hôpitaux Universitaires de Strasbourg, 67000 Strasbourg, France

**Keywords:** *BBS5* gene, Bardet–Biedl syndrome, whole-genome sequencing, structural variation, primary cilium

## Abstract

Bardet–Biedl syndrome (BBS) is an autosomal recessive ciliopathy that affects multiple organs, leading to retinitis pigmentosa, polydactyly, obesity, renal anomalies, cognitive impairment, and hypogonadism. Until now, biallelic pathogenic variants have been identified in at least 24 genes delineating the genetic heterogeneity of BBS. Among those, *BBS5* is a minor contributor to the mutation load and is one of the eight subunits forming the BBSome, a protein complex implied in protein trafficking within the cilia. This study reports on a European *BBS5* patient with a severe BBS phenotype. Genetic analysis was performed using multiple next-generation sequencing (NGS) tests (targeted exome, TES and whole exome, WES), and biallelic pathogenic variants could only be identified using whole-genome sequencing (WGS), including a previously missed large deletion of the first exons. Despite the absence of family samples, the biallelic status of the variants was confirmed. The BBS5 protein’s impact was confirmed on the patient’s cells (presence/absence and size of the cilium) and ciliary function (Sonic Hedgehog pathway). This study highlights the importance of WGS and the challenge of reliable structural variant detection in patients’ genetic explorations as well as functional tests to assess a variant’s pathogenicity.

## 1. Introduction

Bardet–Biedl syndrome (BBS; MIM# 209900) is a rare autosomal recessive ciliopathy affecting 1/160,000 live births in Northern European populations [1] or at a higher rate in highly consanguineous populations [2,3]. The primary clinical features are retinitis pigmentosa (RP), polydactyly, renal anomalies, obesity, cognitive impairment, and hypogonadism with high levels of phenotypic heterogeneity. To date, 24 genes and a few candidates have been identified as responsible for BBS, contributing to high levels of heterogeneity in this disorder [4]. BBS is a part of a group of ciliopathies caused by a dysfunction of the primary cilia. Primary cilia are microtubule-based membrane organelles that protrude apically from the surface of nearly all quiescence mammalian cell types [5,6]. Primary cilia are considered as separate organelles as their membrane is enriched in a unique set of proteins and signaling components [7]. They function as sensory organelles that coordinate a variety of signaling transduction pathways that regulate diverse cellular processes (cell polarity, differentiation, migration, and proliferation) during embryonic development and that maintain tissue homeostasis, such as Hedgehog (Hh), transforming growth factor Beta (TGF-β), and WNT signaling [8]. The vast range of symptoms caused by the defect of the cilium underscores the broad physiological importance of cilium-based signaling.

In vertebrates, Hh signaling is one of the most studied pathways connected and strictly dependent on an intact primary cilium. The Hh pathway is activated when the glycoprotein Sonic Hedgehog (SHH) binds to its receptor PTCH1 (Patched 1), a 12-transmembrane protein. This inactivates PTCH1 and releases SMO (Smoothened), a 7-transmembrane GPCR-like (G protein-coupled receptor) protein [9]. SMO translocates and accumulates into the cilia membrane, leading to the release of GLI proteins from their negative regulator SUFU (SUppressor of FUsed). Activation of the GLI family proteins initiates the Hh downstream signaling cascade [10]. Indeed, the three members of this zinc-finger transcription factors family (GLI1, GLI2, and GLI3) share partially redundant domains and different functions. GLI1 acts as a full-length transcriptional activator, while GLI2 and GLI3 may be subject to post-transcriptional and post-translational processing that will determine their function as either positive or negative regulators. Studies have demonstrated that GLI2 acts mainly as an activator, while GLI3 as a repressor of transcriptional behavior. In response to SHH ligand binding, activated SMO will prevent the proteolytic cleavage of GLI, the full-length activated GLI proteins (predominantly GLI2A) translocate to the nucleus, turn on transcription of target genes, including *Ptch1* and *Gli1* [5], and guarantee proper primary cilium functioning. 

Primary cilium assembly, resorption, and function as a sensory organelle requires the cooperation of several macromolecular machines, one of which is the BBSome. The BBSome is an octameric protein complex consisting of BBS1, BBS2, BBS4, BBS5, BBS7, BBS8 (TTC8), BBS9 (PTHB1), and BBS18 (BBIP1) [11,12]. It is recruited to the ciliary base and is mainly involved in ciliary protein trafficking by delivering protein cargoes transported by IFT trains and in the exit of activated GPCRs and peripheral membrane proteins from the cilium [13]. As the BBSome is involved in the exit of GPCRs from the primary cilium such as PTCH1, SMO, and others, studies suggest that this complex may have an impact on the regulation of the Hh signaling pathway [14]. Among those, BBS5 (NP_689597.1, MIM# 615983) is a 341 amino-acid peripheral sub-unit of the BBSome encoded by 12 exons (NM_129880). Since its discovery in 2004 [15], *BBS5* has been only a minor contributor to BBS, with only 2% of families from various ethnic backgrounds [16]. Comprehensive bioinformatics analysis on BBS5 revealed two novel domains in the BBS5 protein similar to the pleckstrin homology (PH) domains never found in any other BBS protein. These domains can bind to phosphoinositides and allow BBS5 to interact directly with the membrane [11]. This makes it a unique protein both structurally and functionally among BBSome components. Despite these predictions, the functional role of BBS5 and its importance in the BBSome remains poorly understood.

Given that pathogenic variants in BBS genes are known to cause ciliogenesis and Hh signaling anomalies [17], it was interesting to study BBS5 function and the impact of its variants on the structure and function of the primary cilium. This study reports on the genetic analyses undergone by a patient with a severe BBS phenotype. Interestingly, the pathogenic variants could only be revealed by WGS. Specific functional tests revealed the pathogenicity of the variants and shed light on the BBS5 cellular phenotype. Furthermore, the short list of *BBS5* pathogenic alleles was expanded with a novel structural variant and an amino acid duplication.

## 2. Results

### 2.1. Index Patient Case History

Individual AII.2 (the second child in a family of three children) born to non-consanguineous healthy parents with healthy siblings. At birth, a weight of 4.1 kg, height of 51.5 cm, and lower limb polydactyly were noted. Early onset visual impairment caused by pigmentary retinopathy was stated early in childhood with associated obesity, mild intellectual disability (ID), atrophy of the uterus, *situs inversus*, renal agenesis and insufficiency. A clinical diagnosis of Bardet–Biedl syndrome was made. End-stage renal failure, hypertension, and subsequent dialysis was instituted at the age of 40. On clinical examination, weight was 113 kg, having peaked at 240 kg in her late twenties. Visual acuity was reduced to perception of light in both eyes with posterior subcapsular cataract and advanced retinal degeneration. Parents and siblings are not available for biological sampling, so only the DNA of the patient was analyzed with the proband’s fibroblasts. 

### 2.2. Identification and Characterization of Biallelic Variants in BBS5

The patient was referred for in-depth genetic investigations as she remained unsolved following routine genetic diagnostic testing (recurrent *BBS1* and *BBS10* pathogenic variants screening, TES, and WES) [18,19]. This led ultimately to solo whole-genome sequencing since no other family member was available. Out of 4,851,744 SNVs and indels and several thousand structural variants, two interesting variations could be identified in the *BBS5* gene. 

The first variation corresponds to a large heterozygous deletion of 5918 bp (chr2:g.170,333,792_170,339,709del) encompassing the first two exons of *BBS5* (NM_152384.2:c.1-2272_142+879del) (Figure 1A). The variation was confirmed using a specific duplex PCR and breakpoints sequencing performed on genomic DNA (Figure 1B and Appendix A). Interestingly, this large deletion was missed by other NGS assays encompassing the *BBS5* gene, and only WGS identified the deletion (Appendix A). To better understand this, eight copy number callers were applied on the WES and/or WGS (Appendix A). The callers systematically failed to detect the deletion on the WES unless the number of control samples for CANOES was raised (Appendix A). Using the WGS as input data, the detection was much better, with five out of seven programs able to detect the deletion. Investigation of the *BBS5* genomic loci revealed higher GC content in the first exons that prevented its detection using enriched NGS libraries (TES, WES) compared to the more homogeneous depth of coverage observed in WGS data (Appendix A). As the deletion encompassed the promoter, no expression of the allele harboring this variant was expected. To confirm this hypothesis, the RNA expression of *BBS5* extracted from the patient’s skin fibroblasts was evaluated by RTqPCR (Figure 1B). The results showed significantly reduced *BBS5* expression in the patient compared with that in controls, suggesting that only a single allele was expressed. This variant is absent from control populations and is classified as pathogenic (class 5).

The second variation was a heterozygous duplication of one AAT triplet (NM_152384.2:c.550_552dup, p.(Asn184dup)) in exon 7 of *BBS5*. This variant is absent from control populations and is classified a priori as a variant of uncertain significance (class 3) without further investigation. In order to determine the biallelic status of the variants and given the likely non-expressed allele carrier of the deletion of exons 1 and 2, examine the patient’s RNA. It was hypothesized that if both variants were *in trans*, one should observe the AAT duplication at the homozygous state on the Sanger sequence (only the allele carrying the AAT duplication is expressed), or if both variants were *in cis,* one should not be able to detect the duplication at all (the only allele expressed is the one carrying no variant). Sanger sequencing of the patient’s RNA revealed the duplication at the homozygous state, confirming the biallelic status of both variations (Figure 1C) and the fact that the allele harboring the deletion of exons 1 and 2 was not expressed. Localized in the PH2 domain (Figure 2A,B), the asparagine at position 184 is highly conserved across eukaryotes (Figure 2C). Bioinformatics predictions considered this amino acid (AA) duplication deleterious (Appendix A). Interestingly, this position has already been involved (p.Asn184Ser) in a patient with BBS (Appendix A). In order to see whether the Asn184 duplication might affect the secondary structure of the BBS5 protein, PSIPRED [20] was applied to the wildtype and mutated sequence of BBS5 but could not find any significant difference (Appendix A). Localization of the AA 184 in the BBSome 3D structure revealed a buried position within the PH2 domain (Figure 2C). However, this position is localized in a loop between two beta sheets. A Western blot was performed on the patient’s skin-derived fibroblast lysate and three controls using an anti-BBS5 antibody to determine the impact of the duplication at the protein level. The Western blot revealed undetectable levels of BBS5 in the patient’s fibroblasts (Figure 1D), suggesting that the duplication was likely destabilizing the BBS5 protein. This result modified the variant classification from class 3 to class 5 (pathogenic). 

### 2.3. BBS5 Exons 1 and 2 Deletion Screening in a Large Cohort

Owing to the difficulty of detecting exons’ 1 and 2 deletion by standard techniques, it was hypothesized that other unsolved patients with a suspected phenotype related to *BBS5* in our cohort may also carry this deletion. Therefore, duplex PCR was carried out (Appendix A) on 440 inconclusive samples from the cohort including 284 BBS patients and 156 patients with non-syndromic RP. No other patients carrying the deletion could be identified.

In order to determine the frequency of this deletion in the general population, bioinformatics screening was also performed in 863 genomes from a French cohort of healthy individuals (FranceGenRef consortium dataset) for whom visual inspections of the bam files (IGV batch scripting) did not reveal any corresponding alleles. These data are consistent with the screening of the 2504 controls from 1000 Genomes and from the gnomAD SV 2.1 dataset comprising 14,891 controls from various origins. Thus, the found deletion and duplication are two novel extremely rare variants that will be added to the list of 32 pathogenic alleles found so far in *BBS5* (Appendix A).

### 2.4. Non-functional BBS5 Affects Primary Ciliogenesis and Ciliary Length

To analyze primary cilia assembly in cells of the patient harboring bi-allelic *BBS5* variants, ciliogenesis was induced by using serum-starvation-mediated cell-cycle arrest in confluent fibroblasts from the patient and controls and visualized primary cilium by immunostaining with antibodies raised against ARL13B and acetylated γ tubulin for the basal body. To determine the normal state of ciliogenesis, five anonymous control samples were investigated. No significant differences were observed between them, and all had more than 60% of their cells ciliated. The controls were thus pooled for comparison with the BBS patient’s fibroblasts. Immunofluorescence analysis in the patients’ skin-derived fibroblasts revealed that these cells had cilia anomalies. Indeed, when grown in ciliary conditions (0.1% FCS), the patient’s cells developed significantly fewer primary cilia compared to controls (51% vs. 77%) (Figure 3C). Moreover, a significant difference in cilia length could be observed. The patient’s cells harbored cilia significantly longer than those of controls (6–8 µm vs. 3–4 µm) (Figure 3D). *BBS5* variants identified in our patient led to a significant decrease of cilium formation and a longer cilia length. These results indicate that a functional BBS5 protein is important for normal ciliogenesis and cilium length in fibroblast cells. 

### 2.5. Alteration of Canonical Hh Signalling in Patient’s Cells Lacking BBS5

To test the effect of BBS5 on the function of the primary cilium, the Hh pathway’s activity in patient and control cells was questioned using a smoothened agonist (SAG) to activate Hh signaling with and without inducing ciliogenesis (Appendix A). The expression levels of GLI1 and PTCH1 (two Hh target genes) in response to Hh pathway activation were evaluated as a marker of Hh activity. Five controls were used—four of them showed a robust significant induction of Hh target genes GLI1 and PTCH1 in response to SMO ligand stimulation in ciliogenesis conditions (0.1% FCS). By contrast, a strongly reduced response to the SMO ligand was observed in patient cells, revealed by a reduction of the Hh target genes’ expression under the same conditions (Figure 4). As almost no signal was detected in the patient’s fibroblasts, Hh pathway activation was significantly less important in the patient’s cells. These data indicated an alteration of the transduction of the Hh pathway in the patient’s cells that was probably due to the primary cilia alteration caused by the absence of the BBS5 protein.

## 3. Discussion

Bardet–Biedl syndrome is an autosomal recessive ciliopathy characterized by clinical and genetic heterogeneity. Twenty-four BBS genes and a few additional candidate genes have been identified so far, and pathogenic variations are found in a large fraction of them (ranging from 80 to 100%). In such heterogeneous diseases, next-generation sequencing (TES, WES, and WGS) has made the diagnosis for many individuals possible [6,21] but also revealed more complex cases [22,23]. Each of these assays requires a different sample preparation and numerous and diverse bioinformatics programs to analyze the data, revealing sometimes variants by another method, which makes them complementary in solving certain cases. For example, only 25% to 50% of patients have a molecular diagnosis by WES depending on the disease, leaving many cases still unsolved [24]. WGS is now largely used with the advantage of covering the non-coding part of the human genome [25] but requires careful analysis [26] and novel tools [27]. 

In this study, two compound heterozygous variants in *BBS5* were identified*,* a duplication of the triplet AAT in exon 7 (NM_152384.2:c.550_552dup) leading to the duplication of asparagine 184 (p.Asn184dup) and a large deletion of the 2 first exons (c.1-2272_142+879del) encompassing the promotor region of the gene. The patient’s exploration included TES, WES, and finally WGS. The Asn184 duplication was identified by all three methods but was initially overlooked (class 3) with the absence of a second variant, although manual inspection of the TES and WES data using IGV was performed to search for a possible CNV. Previous studies have already shown that certain types of variations might not be detected by some approaches (TES vs. WES vs. WGS) [28,29]. This was the case for the 5918 bp deletion that was only detected by WGS in a region less prone to TES and WES. This shows how WGS is an appropriate strategy for the exploration of patients with rare diseases and that the nature of the collected data (i.e., homogeneous depth of coverage compared to WES) can improve detection of variations even when using the same bioinformatics program.

*BBS5* is a minor contributor to the BBS mutation load as only 2% of the families from various ethnic backgrounds harbor *BBS5* variants [16]. So far, 32 variants have been reported, including missense, nonsense, splicing, deletion or insertion, and some structural variants (Appendix A). Analysis of the clinical data of previously reported *BBS5* cases (*n* = 65 for 45 families) highlights high variability in the severity the patients experience (Appendix A). Among the 39 cases with clinical information, obesity was the most prominent clinical sign (*n* = 29), followed by retinal dystrophy (*n* = 23), polydactyly (*n* = 19), cognitive impairment (*n* = 18), hypogonadism (*n* = 17), and renal anomalies (*n* = 11). Considering the combination of major clinical manifestations, nine had one major sign, five had two, nine had three, eight had four, five had five, and three had six, delineating the variability in clinical manifestations of *BBS5* cases. 

Trio analysis using either WES or WGS generally has a rate of diagnostic success higher than that of singleton analysis [30,31]. Nevertheless, thanks to cDNA analysis that proved that only one allele was expressed, it was possible to demonstrate the biallelic status of variations. Since the parents’ samples were not available, it was not possible to rule out that one or even both variants were of de novo appearance. However, this was less likely given the extremely rare report of such cases [32,33], although underestimated in ciliopathies (Gouronc et al., manuscript accepted).

Given the difficulty in identifying this variation and the genomic context surrounding the breakpoints, it was speculated that other genetic screenings might have missed this variation and that several other families might carry it. However, despite screening 440 samples, no other cases harboring this deletion could be identified as underlying the burden of private pathogenic variations in BBS [18]. The AAT duplication was initially considered as a variant of uncertain significance. Indeed, non-frameshifting indels are a type of variant that are difficult to handle as only a few programs and cases are available to reliably predict any effect. Asn 184 is located in the second DM16 region within the PH2 domain of the protein. This domain’s function is still unknown, but the sequence is very well conserved in evolution [15]. This high level of conservation and the bioinformatics predictions (e.g., deleterious) suggest that the variant should be considered pathogenic. Analysis of the proteins extracted from the cultured skin-derived fibroblasts of the patient did not succeed in revealing any BBS5 protein compared to controls. It was hypothesized that the protein was not stable enough to maintain its 3D structure with a single AA in addition. 

Multiple studies have already shown the implication of BBS genes in regulating ciliogenesis and cilium length with inconsistent conclusions due to various conditions and models [34,35]. In particular, BBS5 is a protein of 341 amino acids (NP_689597.1) and also one of the eight subunits forming the BBSome complex [16]. The BBSome is a 438 KDa complex located at the base of the cilium. It is required for ciliary localization, assembly, and function, as it functions as a cargo adaptor that maintains, stabilizes, and assembles IFT particles; thus, it supports protein and signaling receptors’ trafficking across the cilium [11,36,37]. As the BBSome assists traffic at the base of the cilium, pathogenic variations in BBS proteins might disrupt cilia assembly and cilia function. BBS5 is the only component of the BBSome that contains two PH domains, making its structure unique. These PH domains bind to phosphoinositides, which is critical for ciliogenesis. Even if its function is not well understood, BBS5 was recently associated with ciliogenesis defects (less ciliated cells and shorter cells) and the regulation of the transduction of the Hh signaling pathway [38]. In this study, similar results were obtained, strengthening the observed effect. On the contrary, a similar investigation on different models or cell types showed mixed effects. Depletion of the *BBS5* gene in retinal pigment epithelial (RPE) cells shows a reduction in the number of ciliated cells [12]. However, other studies demonstrated no effect such as in a BBS5 knockdown (KO) in mouse embryonic fibroblast cells (MEFs cells) [39] or in hTERT-RPE-1 knockdown cells [40]. These differences might indicate different effects depending either on the cell type or on the pathogenic variants in consideration. Demonstrating consistency in how BBS genes affect the patients’ cells is important to assess future variants of uncertain significance.

It is well-established that a functional primary cilium is particularly important for the development of the body through the activation of signaling pathways such as the one mediated by SHH [41]. In the same way, investigation of the Hh signaling pathway showed that a defective BBS5 leads to inactivation of the pathway. Another study observed similar results in other *BBS5* patient’s fibroblasts and in *BBS5*-RPE1 KO cells [38]. On the contrary, *BBS5^−/−^* mouse models do not exhibit classical Hh signaling defects (e.g., dorsal ventral neural tube patterning defects, polydactyly). Still, they identified developmental and homeostatic effects and novel pituitary abnormalities [39]. The observed impairment of the Hh signaling activity due to *BBS5* variants or *BBS5*-depletion shows the importance of BBS5 function in the BBSome and its implication in the trafficking of receptors important for the canonical Hh signaling pathway. It is not certain that the different BBS proteins in the BBSome are equally crucial for Hh signaling; they may have protein-specific functions. Alteration of the Hh pathway has been associated with polydactyly in several studies [42]. This is indeed a typical phenotype present in patient AII.2. Results such as these call for a more systematic analysis of cells for various conditions including different types of variants and different genes. To understand if the BBS5 effect on the Hh pathway is cell-type-specific or species-specific, further studies on different cell lines and *BBS5* mice models are needed. 

Accessing a definitive molecular diagnostics in the case of rare genetic diseases is essential for providing proper genetic counselling to the affected family and access to existing treatments [43,44]. Reporting additional cases associated with this gene should help identifying genotype–phenotype correlations and lead to novel clinical trials in the future.

## 4. Materials and Methods

### 4.1. Molecular Genetics Investigation

Several genetics assays were used to investigate the affected member from family A (II.2), including TES of BBS genes and WES (see Appendix A). In particular, WGS was performed for the index case (II.2) by the Centre National de Recherche en Génomique Humaine (Institut de Génomique, CEA). Genomic DNA was used to prepare a library for WGS using an Illumina TruSeq DNA PCR-Free Library Preparation Kit according to the manufacturer’s instructions. After normalization and quality control, qualified libraries were sequenced on a HiSeqX5 platform from Illumina (Illumina Inc., San Diego, CA, USA) as paired-end 150 bp reads. Sequence quality parameters were assessed throughout the sequencing run, and standard bioinformatics analysis of the sequencing data was based on the Illumina pipeline to generate FASTQ files for each sample. The sequence reads were aligned to the reference sequence of the human genome (GRCh37) using the Burrows–Wheeler Aligner (BWA V7.12) [45]. The UnifiedGenotyper and HaplotypeCaller modules of the Genome Analysis ToolKit (GATK) [46], Platypus [47], and Samtools [48] were used for calling both single nucleotide variations (SNV) and small insertions/deletions (indels). An average sequencing depth of 37× was achieved for 91.6% of the genome. Annotation and ranking of SNVs and indels were performed by VaRank 1.4.3 [49] in combination with Alamut Batch software 1.11 (Interactive Biosoftware, Rouen, France). Very stringent criteria were applied to filter out non-pathogenic variants (Appendix A): (i) variants represented with an allele frequency of more than 1% in public variation databases, including 1000 Genomes [50], the gnomAD database [51], the DGV database [52], or our internal database; (ii) variants in 5′ and 3′ UTR, downstream, upstream, or intronic locations and synonymous without pathogenic prediction of local splice effects; (iii) variants not in the ciliary genes [53]. Structural variants were predicted using CANOES with a bed restricted to the WES [54] and Lumpy 0.2.13 [55] including mobile element insertion using Mobster [56] and annotated by AnnotSV 2.3 [57]. The analysis focused on compound heterozygous and homozygous variants consistent with a recessive mode of transmission. Each candidate variation was also checked using Integrative Genomics Viewer (IGV 2.16.0) software [58].

### 4.2. Cell Culture

Primary dermal derived fibroblasts from a BBS case (II.2) and control individuals were obtained by skin biopsy according to [59]. The fibroblasts were cultured in Dulbecco’s modified Eagle’s medium (DMEM) with glutamax (21885-025 Gibco, Thermo Fisher Scientific, Waltham, MA, USA) supplemented with 10% fetal calf serum (FCS) and 1% penicillin–streptomycin (PS) (all from Gibco). To induce ciliogenesis, the fibroblasts were grown in serum-reduced media (0.1% FCS) for 48 h. To induce SHH pathway stimulation, the fibroblasts were stimulated with smoothened agonist (SAG, ab145866 Abcam, Cambridge, UK) at a final concentration of 100 nM for 48 h with and without serum starvation.

### 4.3. Genomic DNA Extraction, PCRs

DNA was extracted from blood cells using a DNeasy Blood & Tissue Kit (ref 69504, Qiagen, Hilden, Germany). DNA concentration was evaluated by spectrophotometry (Eppendorf Bio Photometer Plus). Genotyping PCRs were carried out with diluted DNA to 10 ng/reaction, diluted Tag DNA polymerase (2U/reaction), and PCR mix 5× to 1×. A duplex PCR was designed to detect the deletion specifically. PCR protocol variations were applied to optimize the amplification of the fragments of interest (PCR touchdown). Under these conditions, the duplex PCR was carried out using two pairs of primers. The confirmation of the duplication was carried out by a classic PCR on cDNA. PCR was carried out using 10 ng of cDNA/reaction. Primers for both PCRs were designed using Ensembl [60], Primer3 “https://github.com/primer3-org” (accessed on 1 February 2020), the UCSC “in silico PCR” [61], and NCBI primers [62]. Primers are available in Appendix A. PCR conditions are available upon request.

### 4.4. Sanger Sequencing and Segregation 

Variant confirmation or cDNA sequences were obtained using Sanger sequencing after PCR amplification and fragments’ purification (membrane placed under vacuum for 10 min). Bidirectional sequencing of the purified PCR products (with the appropriate primers) was performed by GATC Sequencing Facilities (Konstanz, Germany). The used primers are summarized in Appendix A. 

### 4.5. Cohort Screening

Two duplex PCRs were designed to detect specifically the deletion of exons 1 and 2. Primers and example conditions are provided in Appendix A. 

### 4.6. RNA Extraction, Reverse-Transcription, and Quantitative Real-Time PCR 

RNA was extracted from skin-derived fibroblasts via QiaShredder (Qiagen, ref 79654) and RNeasy Minikit (Qiagen, ref 74104) kits. RNA concentration was evaluated by spectrophotometry (Eppendorf Bio Photometer Plus). cDNA was synthesized via an “iScript ™ c Synthesis Kit” (170-889, Bio-Rad, Hercules, USA) according to the manufacturer’s instructions with 500 ng of RNA. Quantitative real-time PCR was carried out with a iQTM SYBR Green Supermix (#170-8886, Bio-Rad) in a Bio-Rad CFX96/384^TM^ Real-Time System in triplicate. cDNA was diluted to 1/10, and *BBS5*, *GLI1*, *PTCH1*, *GAPDH,* and *HPRT* (two house-keeping genes) primers were used at a concentration of 0.25 µM. The primers are available in Appendix A. The normalized fold expression of the target genes was calculated using the comparative cycle threshold (*C*_t_) method (2^−ΔΔCt^ method) by normalizing target messenger RNA (mRNA) *C*_t_ to those for both *GAPDH* and *HPRT* reference gene using CFX Manager Software Version 1.5 and Microsoft Excel calculations. Data are presented as relative expressions ± SEM. Statistical analyses were done using the GraphPad Prism software. *p*-values and significance were determined using a two-way ANOVA test (Tukey’s multiple comparison test). ** *p* < 0.01 **** *p* < 0.0001. 

### 4.7. Protein Extraction and Western Blot 

Protein was extracted from skin-derived fibroblasts with a radioimmunoprecipitation assay buffer (RIPA buffer) containing Protease Inhibitor Cocktail (25×, Complete EDTA free, 05056489001, Roche, Basel, Switzerland). Protein quantification was performed with the Bradford test using an Eppendorf BioPhotometer Plus spectrophotometer. A total of 40 μg of protein was separated on 10% SDS-PAGE polyacrylamide gel (Mini-PROTEAN TGXTM Gels cat.#456-1034, Bio-Rad) or homemade gel and transferred onto a PVDF membrane (0.2 μM, Trans-Blot Turbo, Transfer Pack, Bio-Rad, 1704156) using the TransblotR TurboTM Transfer System (Bio-Rad). After blocking in 5% milk–TBS1X–Tween 0.1% for 1 h at room temperature, the membrane was probed with the primary antibody overnight at 4 °C, followed by HRP-conjugated secondary antibody incubation for 1 h at room temperature. Chemiluminescent detection was performed using a ChemidocTM MP Imaging System (Bio-Rad) with PierceTM ECL Western (ref 32209, Thermo Fisher Scientific, Waltham, MA, USA) and SuperSignal R West Femto Maximum Sensitivity Substrate (prod #34095, Thermo Fisher Scientific) solutions. Stripping was done using the RestoreTMWestern Blot Stripping BufferTM (prod #21059). Quantification of bands was performed using Image Lab software 6.0.1 (Bio-Rad). BBS5 quantification data performed in triplicate were normalized to the reference cytoplasmic protein β-Tubulin. Antibodies used in this experiment are available in Appendix A.

### 4.8. Immunofluorescence Analysis

To induce primary cilia formation, the fibroblasts were grown in serum-reduced media (0.1% FCS) for 48 h. Skin-derived fibroblasts were grown on 24-well plates (Falcon, 3226) and Lab-Tek8-well chamber slides (Thermo Fisher Scientific Nunc 15411). The fibroblasts were subsequently fixed in ice-cold methanol at −20 °C for 10 min, permeabilized with 0.1% Triton X-100 in PBS 1×, and blocked in 10% FCS in PBS 1× for 1 h at room temperature. Incubation with primary antibodies diluted in 2% FCS was done for 1 h at room temperature or overnight at 4 °C. Secondary antibody incubation was carried out for 1 h at room temperature. For those realized in Lab-Tek8-well, nuclei were visualized using Hoechst, and images were obtained using an inverted fluorescence microscope (Olympus spinning disk, 60× water immersion objective). For those realized on 24-well plates, images were obtained using an IncuCyte S3-Sartorius and were analyzed using ImageJ. Cilia frequency was calculated from the number of cells with a primary cilium divided by the total number of cells (*n* = 400 cells analyzed). *p*-values and significance were determined using ordinary one-way ANOVA ****: *p* < 0.0001. Primary cilia length was measured via IncuCyte software irrespective of the angle of orientation (*n* = 400 cilia). *p*-values and significance were determined using Kruskal–Wallis test ****: *p* < 0.0001. The number of cilia (*n*) measured consisted of pooled data from two separate experiments performed on biological triplicates of each sample in each experiment. Statistical analysis was done using GraphPad Prism software. Antibodies used in this experiment are available in Appendix A.

## 5. Conclusions

In this study, two novel variations in *BBS5* (c.550_552dup, and c.1-2272_142+879del) were identified in a non-consanguineous European BBS patient that broadened the *BBS5* mutation spectrum. The pathogenicity of both variations was confirmed. The duplication likely destabilizes the protein and is now classified as a class 5 variant. Furthermore, it was demonstrated that BBS5 affected ciliogenesis and primary cilium length. The Hh signaling pathway was impaired in the patient’s fibroblasts, indicating a defective primary cilium and that BBS5 also affected primary cilia function. This report illustrated the usefulness of WGS sequencing in identifying pathogenic variations, especially CNVs in highly heterogeneous genetic disorders. Despite robust ACMG guidelines for variant interpretation, this study shows how functional tests are essential to providing a definitive answer to patients.

## Figures and Tables

**Figure 1 ijms-24-08729-f001:**
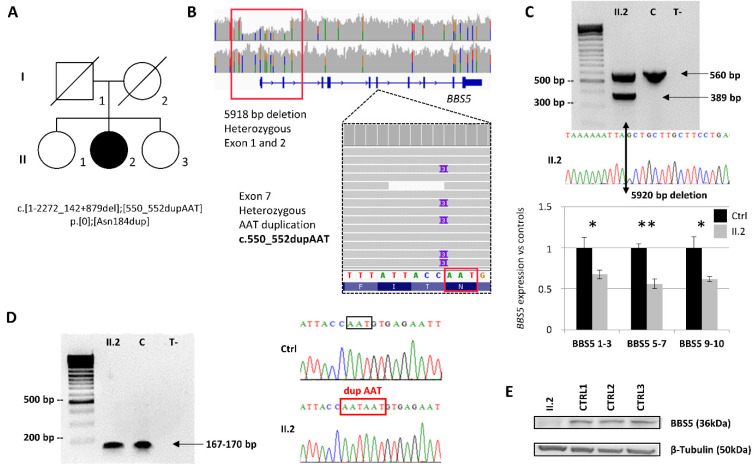
Identification of two variants in *BBS5* for a family explored by WGS. (**A**) Pedigree of the family including one affected daughter (II.2). (**B**) IGV visualization of the two identified pathogenic variants; a large heterozygous deletion overlapping exons 1 and 2 (upper panel) and the heterozygous AAT duplication in exon 7 (lower panel). (**C**) Large deletion analysis and confirmation including Sanger sequencing of the breakpoint and a duplex PCR (genomic DNA). Quantitative real-time PCR was performed on RNA extracted from fibroblasts of the patient and a healthy unrelated control, using three pairs of primers amplifying between exons 1–3, exons 5–7, and exons 9–10. *BBS5* expression data were performed in triplicate. The normalized fold expression of the target genes was calculated using the comparative cycle threshold (*C*_t_) method (2^−ΔΔCt^ method) by normalizing target mRNA *C*_t_ to those of two reference genes (*GAPDH* and *HPRT*). Data are presented as relative expressions ± SEM (ANOVA, *: *p* < 0.1, ** *p* < 0.01). (**D**) AAT duplication confirmation and segregation analysis. PCR and Sanger sequencing were performed on cDNA to confirm the duplication and to assess the biallelic status of the variants. The AAT duplication appeared homozygous on the sole expressed allele. (**E**) BBS5 protein expression in skin-derived fibroblasts was revealed by Western blot using an anti-BBS5 antibody.

**Figure 2 ijms-24-08729-f002:**
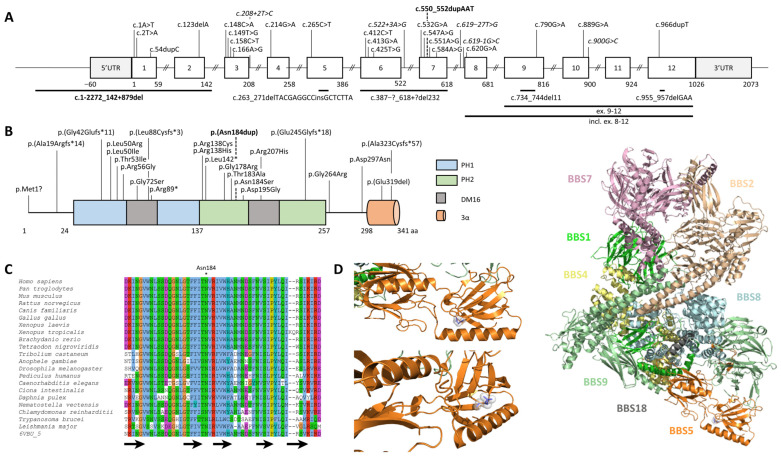
*BBS5* gene and protein structure. (**A**) Schematic view of the *BBS5* gene and (**B**) protein together with known pathogenic variations. The two variations from family AII.2 are indicated in bold, and variations with an effect on splicing are in italic. Large deletions are displayed as horizontal lines. (**C**) Multiple sequence alignments of BBS5 proteins from different metazoan species showing high conservation of the amino acid at position 184. Asn184 position is highlighted by a “*”. Black arrows indicate β-sheets. (**D**) Structure of BBSome complex indicating the positioning of BBS5 regarding the other subunits (right part). BBSome subunits have been colored according to the corresponding scheme: BBS1 (pale blue), BBS2 (wheat), BBS4 (pale yellow), BBS7 (light pink), BBS8 (alecyan), BBS9 (pale green), BBS18 (gray), and BBS5 is shown in orange (PDB: 6VBU). Specific focus on the AA at position 184 (shown in blue sticks/spheres tone) localizing the AA in a loop within the second PH domain (left part).

**Figure 3 ijms-24-08729-f003:**
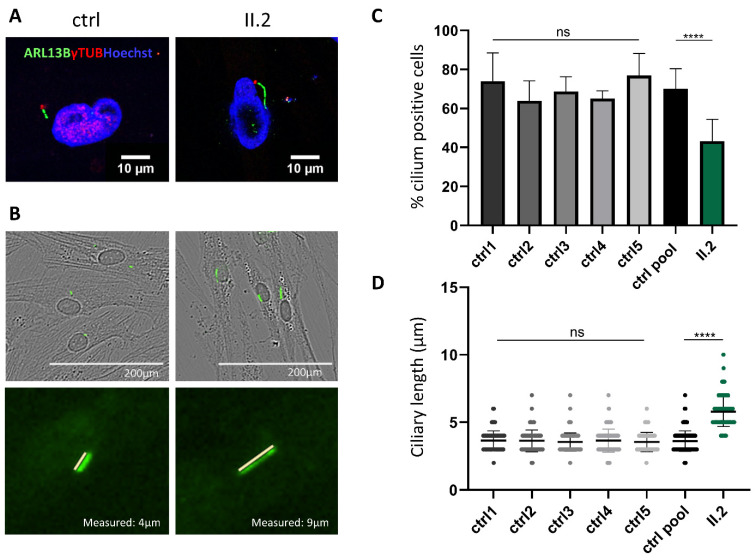
Functional assessment of *BBS5* variations. (**A**) Representative images of ciliogenesis in skin-derived fibroblasts from controls and the patient. Primary cilia were labeled with an anti-ARL13B antibody (green), the basal body with anti-γ-Tubulin antibody (red), and the nucleus was visualized with Hoechst staining (blue). Images were taken with an inverted fluorescence microscope (Olympus spinning disk, 60× water immersion objective). Scale bars 10 µm. (**B**) Ciliated cells were counted on serum-deprived (48 h) controls and the patient’s skin-derived fibroblasts fixed for immunofluorescence and stained with an anti-ARL13B (green) antibody. Images were taken with an Incucyte S3 Sartorius (20× objective), allowing a combined visualization of phase (cells) and fluorescence (cilia). (**C**) Percentage of patient’s skin fibroblasts with cilia. The percentage of ciliated cells was calculated by dividing PC counts by cellular counts (*n* = 400 cells analyzed for each of the controls and the patient). *p*-values and significance were determined using ordinary one-way ANOVA, ns: non-significant, ****: *p* < 0.0001. (**D**) Quantification of primary cilia length (*n* = 400 cells analyzed for each of the controls and patient). *p*-values and significance were determined using the Kruskal–Wallis test ****: *p* < 0.0001. In (**C**,**D**), the measured number of cilia (n) represents pooled data from two separate experiments performed on biological triplicates of each sample in each experiment.

**Figure 4 ijms-24-08729-f004:**
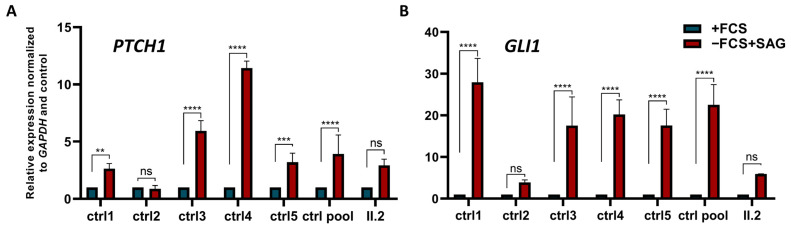
Assessment of Hh-signal transduction by quantification of the expression levels on Hh-stimulated cells (+SAG) from controls and the patient with (−FCS) and without (+FCS) inducing ciliogenesis. Results for two Hh target genes are represented (**A**) *PTCH1* and (**B**) *GLI1.* The expression of both genes was normally induced in controls except in control 2, whereas no induction was found in the patient. The expression data performed with biological triplicates were normalized to the reference gene *GAPDH* mRNA expression data, and the values were presented as relative expression levels ± SEM. *p*-values and significance were determined using a two-way ANOVA test (Tukey’s multiple comparison test). ns: non-significant, **: *p* < 0.01, ***: *p* < 0.001 ****: *p* < 0.0001.

## Data Availability

Data generated or analyzed during this study are included in the published article and the corresponding Appendix A. The raw sequencing data generated in the course of this study are not publicly available due to the protocol and the corresponding consent used that did not include such information. All variants have been submitted to ClinVar (https://www.ncbi.nlm.nih.gov/clinvar/) and are accessible using the following accessions numbers: SCV003838996, SCV003838997.

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
