# Peer review of "WGS Revealed Novel *BBS5* Pathogenic Variants, Missed by WES, Causing Ciliary Structure and Function Defects"

_ijms, 2023, doi:10.3390/ijms24108729_

Round 1
Reviewer 1 Report
Overall, the study provides a clear and concise summary of the study on Bardet-Biedl syndrome (BBS) with a focus on the identification of biallelic pathogenic variants in BBS5 in a European family with severe BBS phenotype. The study provides an overview of the genetic heterogeneity of BBS, the role of BBS5 in protein trafficking within the cilia, and the use of multiple next-generation sequencing tests, including whole-genome sequencing, to identify the pathogenic variants. The study also highlights the challenge of reliable structural variant detection in patients' genetic explorations and the importance of functional tests to assess variant pathogenicity.
Author Response
We thank the reviewer for the kind words and the perfect understanding of this work.
Reviewer 2 Report
Karam and coworkers describe novel BBS5 mutations in a patient affected by a Bardet-Biedl syndrome.
The paper is of interest and provides further evidence on the role of BBS5 in ciliary function.
The WGS approach was applied to identify the pathogenic variants and patient’s fibroblasts used to prove the consequence of BBS5 mutation on ciliary structure.
Authors should take into consideration the following suggestions:
1) Lines 2-3: the title does not seem to appropriately reflect the content of the study. I wonder if functional ciliary effects can be revealed by WGS. WGS can reveal sequence variants. Moreover, immunocytochemical and gene expression analyses demonstrate altered ciliogenesis “possibly” affecting function.
2) Line 29: the paper does not describe a family, but only the patient since no family samples were available.
3) Line 34: the term “cell morphology” is not appropriate since the authors describe the presence/absence of cilia and the size of cilia, they do not describe the morphology of the cells.
4) Lines 148-152: the sentence is not clear. Authors should better clarify how duplication can be demonstrated.
5) Line 163: It is not clear how BBS5 antibody can demonstrate the impact of duplication at protein level. Protein immunodetection in patient’s fibroblasts is the result of both sequence variations.
6) Line 399: Authors should specify the age of donors and the site where fibroblasts come from. Since data provided in figure 4 are quite heterogeneous, I wonder if these cells are from comparable sites and from subjects of comparable age.
Only minor editing is required
Reviewer 3 Report
Adella Karam et al submitted an article “Structural and functional ciliary effects of novel variants in BBS5 revealed only by WGS”.
The clinical and molecular data very are very interesting and could be of interest to the readers. However, there are some comments that need to be addressed.
General comments:
+ Follow nomenclature: https://varnomen.hgvs.org/
*- The reference sequence (NP_) and NM_ should also be added.
**Gene name should be in italics. Kindly check.
**USE (OMIM #), and use the proper name of the disorder.
--Spacing issues need attention.
++English grammar should be improved.
Introduction:
Kindly cite recent literature.
Methods/Results
**IRB approval number, university name, consent approval should be mentioned in the first paragraph.
Results
How the variants were classified as pathogenic ACMG criteria fulfilled? HOW VUS were justified?
**Was the patient treated with any medication? Condition of patients before and after treatment?
Use 3 letter code for Amino acids in the mutation nomenclature.
Any unique clinical description among the affected individual? Facial anomalies or any other medical condition? Comparison with other cases from literature is missing.
Make a list of all the variants identified for the gene and perform genotype-phenotype correlations.
Discussion
Any genotype-phenotype correlation? Location of variants associated with variable phenotypes? Hotspot variant?
WES as a diagnostic tool and discovery of novel genes due to the advent of WES and supporting technologies should be discussed. PMID: 35709191, PMID: 31656313, PMID: 30498240
In the last discussion, add lines for future perspectives; discuss newborn screening, NIPT, PGT-A, and PGT-M, for example.:
· Proper genetic counseling for the affected family is essential in the case of rare genetic diseases. Furthermore, parenteral genetic screening/diagnosis is the best strategy for managing this disease, which currently has no therapy (PMID: 31557427; PMID: 33613643; PMID: 33804821). Reporting additional cases associated with this gene would help identify genotype–phenotype correlations and lead to clinical trials in the future (PMID: 36406136; PMID: 34635114).
Rare genetic disease prevention strategies and the future of gene therapy should be discussed.
References missing:
· PMID: 28761321; PMID: 29232001
